# Nickel-catalyzed switchable 1,3-dienylation and enantioselective allenylation of phosphine oxides

Jiayin Zhang[1], Xihao Chang[1], Xianghong Xu[1], Hongyi Wang[1], Lingzi Peng[1] & Chang Guo [1]✉

The development of general catalytic methods for the regio- and stereo-selective construction of phosphoryl derivatives from identical substrates remains a formidable challenge in organic synthesis. Enabled by the newly developed BDPP-type ligands, we disclosed a nickel-catalyzed allenylation of phosphine oxides rationally and predictably, allowing the construction of versatile chiral allenylphosphoryl derivatives with high enantiopurity (up to 94% e.e.). Alternatively, using an achiral phosphine ligand dcypbz under acidic conditions, we achieved a regiochemical switch of the 1,3-dienylation to afford functionalized phosphinoyl 1,3-butadienes (up to 93% yield). The salient features of this method include switchable reactivity, broad substrate scope, readily available feedstock, single-step preparation, and high asymmetric induction.

Optically active allene-containing organic compounds have found widespread application in organic synthesis and materials chemistry[1–3]. A synthetic method that can rapidly lead to chiral substituted allenes from simple and readily available starting materials is highly desirable[4–8]. Concerning efficiency and versatility, transition metal catalysis has emerged as an attractive method to transform propargylic alcohol derivatives into propargylated[9–19], 1,3-dienyl[20,21], or allenyl products[22–31] through the use of various nucleophiles[32–34]. However, control of regioselectivity in catalytic propargylic substitution reactions has rarely been demonstrated[35,36]. We hypothesized that it might be possible to tune the desired selectivities through careful choice of phosphine ligand, leading to the formation of phosphoryl derivatives with functional diversity.

Allenylphosphine oxides and phosphinoyl 1,3-butadienes are widely recognized as synthetic intermediates[37–39], chiral ligands[40,41], and biologically active reagents[42]. Therefore, the development of enantioselective methods to access diverse allenylphosphoryl derivatives rapidly and efficiently is highly desirable. Classical synthetic avenues toward allenylphosphine oxide synthesis usually rely on [2,3]-sigmatropic rearrangement[43,44]. The process hitherto developed still suffers from severe limitations, such as multistep synthetic sequences,

the use of relatively unstable and hazardous phosphorus chlorides, and poor tolerance of functional groups. Enantioselective couplings of propargylic alcohol derivatives with various nucleophiles have been reported to afford chiral allenes (Fig. 1A)[45–50]. Despite extensive efforts, there is no extant catalytic protocol for constructing enantiomerically enriched allenylphosphoryl derivatives from two readily available fragments. On the other hand, transition metal-catalyzed propargylic substitution reactions have been established for carbon-phosphorus bond formation in racemic versions[51–56]. Recently, Sakata, Nishibayashi, and coworkers described the chiral ruthenium complex-catalyzed enantioselective phosphinylation of terminal propargylic alcohols delivering propargylic derivatives with high regio- and enantioselectivity (Fig. 1B)[57]. The Murakami group reported a seminal nickel-catalyzed reaction of propargylic carbonates with phenols for the synthesis of aryloxy-1,3-dienes[21]. Therefore, the development of a simplified and stereospecific strategy to access various phosphoryl derivatives would be warranted to further widen the synthetic potential of these scaffolds and build up molecular complexity. However, several challenges have to be overcome, such as (1) the selective generation of different products from identical substrates; (2) catalytic regioselective protocols utilizing ligands rather than substrate control;

[1]Hefei National Laboratory for Physical Sciences at the Microscale, University of Science and Technology of China, Hefei 230026, China.
✉e-mail: guochang@ustc.edu.cn

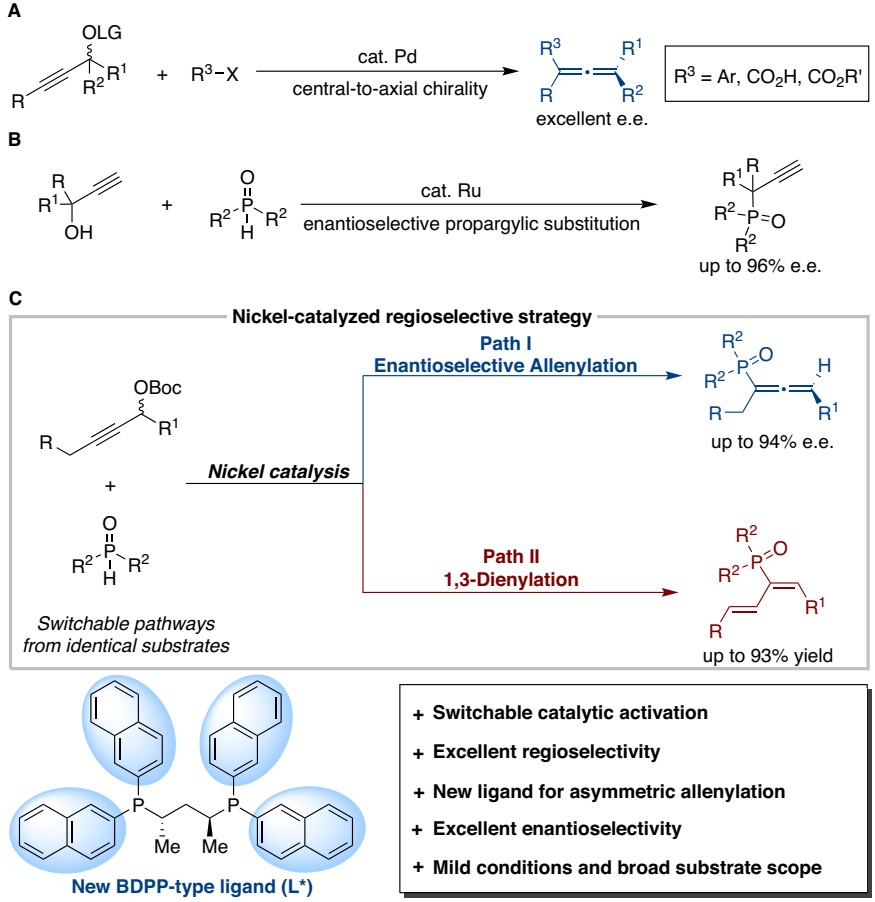

**Fig. 1 | Nickel-catalyzed regioselective strategy for 1,3-dienylation and enantioselective allenylation.** **A** Pd-catalyzed enantioselective allenylation. **B** Ru-catalyzed enantioselective propargylic phosphinylation. **C** This work: Ni-catalyzed 1,3-dienylation and enantioselective allenylation.

and (3) the identification of reaction conditions that afford high enantioselectivities.

The allenylnickel complex, as revealed by recent elegant reports, was proposed as the key intermediate in nickel-catalyzed asymmetric propargylic substitution reactions[58–62]. Therefore, we speculated that the nickel-catalytic system with the rational design and development of chiral ligands would meet the challenges above to achieve carbonphosphorus bond formation in an asymmetric manner. In this work, we report the successful introduction of nickel-catalyzed 1,3-dienylation and enantioselective allenylation, providing phosphinoyl 1,3-butadienes and allenylphosphine oxides with excellent chemo- and regioselectivity (Fig. 1C).

## Results

### Reaction optimization

In an initial experiment, the coupling reaction of racemic propargylic carbonate **1a** and diphenylphosphine oxide **2a** was examined (Table 1). The use of (*S,S*)-Ph-BPE (**L1**) afforded trisubstituted allenylphosphine oxide **3a** and phosphinoyl 1,3-butadiene **4a** as a mixture in a ratio of 1:2 with 26% yield, and **3a** was obtained with 12% e.e. (entry 1). We next evaluated chiral phosphine ligands that were crucial in terms of both reactivity and selectivity (entries 2–8). Gratifyingly, using phosphine ligand *i*Pr-DuPhos (**L3**), 1,3-dienyl isomer **4a** was isolated in 45% yield as the sole product (entry 3). We next evaluated a series of chiral bidentate phosphine ligands (**L4**-**L8**), and a change from *i*Pr-DuPhos (**L3**) to (*S,S*)-BDPP (**L8**) was accompanied by a switch in selectivity from the 1,3-dienyl isomer to the allenyl isomer under otherwise identical reaction conditions (entries 3 versus 8). Further studies focused on the effect of phosphine ligand on

the regioselective reaction (entries 9 and 10), and the use of an achiral dcypbz ligand (**L10**) afforded **3a** and **4a** as a mixture in a ratio of 1:3. Notably, the yield of 1,3-dienyl isomer **4a** increased to 65% with excellent regioselectivity when DMF was employed as the solvent with **L3** as the ligand (entry 11). Interestingly, **4a** can also be accessed as the main product by using diphenylphosphinic acid as the additive with the dcypbz ligand **L10** (entry 12, see Supplementary Information for details). Using **L8** as the chiral ligand, **3a** was delivered in 34% yield and 48% e.e. when the reaction was performed at 25 °C with the use of quinuclidine as an additive (entry 13). The stereoselectivity was further enhanced by using DCM as the solvent (entry 14, 35% yield, 69% e.e.).

Subsequently, we designed and synthesized a series of BDPP analogs (**L11**-**L26**) to alter the electronic and steric properties and further investigate their influence on the reaction outcomes (Fig. 2)[63]. It appears that substituents at the *para* position of the benzene ring linked to the phosphorus atom had some effect on the catalytic properties (**L11**-**L14**), and the stereoselectivity was enhanced to 74% e.e. by using **L13** with the *tert*-butyl group. The steric properties of different substitutions at *meta* positions (**L15**-**L18**) showed no improvement in reaction efficiency. Other multiple substituents were then studied (**L19**-**L22**), and (*S,S*)-3,5-(Me)₂BDPP (**L19**) afforded **3a** with 76% e.e., albeit with low conversion. Significantly, the desired product **3a** could be obtained with 80% e.e. by using the newly synthesized (*S,S*)-(2-Naph)BDPP (**L23**). More importantly, BDPP-type ligand **L23** could be easily prepared on a gram scale from commercially available starting materials (see Supplementary Information for details). The molecular structure and absolute configuration of **L23** were identified by single-crystal X-ray diffraction analysis. No improvement in yield or

## Table 1 | Condition evaluation

| Entry | L | Solvent | 3a/4a | Yield (%)[a] | e.e. of 3a (%) |
|---|---|---|---|---|---|
| 1 | L1 | dioxane | 1:2 | 26 | 12 |
| 2 | L2 | dioxane | 1:3 | 20 | 28 |
| 3 | L3 | dioxane | <1:20 | 45 | - |
| 4 | L4 | dioxane | 3:1 | 45 | 28 |
| 5 | L5 | dioxane | 7:1 | 51 | 16 |
| 6 | L6 | dioxane | >20:1 | 52 | 18 |
| 7 | L7 | dioxane | >20:1 | 51 | 24 |
| 8 | L8 | dioxane | >20:1 | 52 | 43 |
| 9 | L9 | dioxane | 3:1 | 31 | - |
| 10 | L10 | dioxane | 1:3 | 42 | - |
| 11[b] | L3 | DMF | <1:20 | 65 | - |
| 12[c] | L10 | DMF | <1:20 | 78 | - |
| 13[d] | L8 | dioxane | >20:1 | 34 | 48 |
| 14[d] | L8 | DCM | >20:1 | 35 | 69 |

Reactions were conducted by using Ni(COD)$_2$ (10 mol%), L (12 mol%), (rac)–1a (0.3 mmol), and 2a (0.1 mmol) at 80 °C, 24 h. e.e. values were determined by high-performance liquid chromatography analysis.
[a]Isolated yield of the mixture 3a and 4a after chromatography are shown.
[b](rac)-1a (0.12 mmol), 2a (0.1 mmol), Ni(COD)$_2$ (5 mol%), and L3 (6 mol%) was used.
[c](rac)-1a (0.12 mmol), 2a (0.1 mmol), Ni(COD)$_2$ (10 mol%), L10 (12 mol%), and Ph$_2$P(O)OH (0.5 equiv) was used at 100 °C.
[d]Quinuclidine (1.0 equiv) was used at 25 °C. DMF = N,N-dimethylformamide, DCM = dichloromethane.

enantioselectivity was observed with the use of chiral ligand L24. Further optimization studies revealed that the chiral backbone of the phosphine ligands (L25 and L26) showed remarkable effects on the outcome of the reaction, and finally, L23 was identified as the optimal choice.

Notably, the use of HCO$_2$Li as an additive can dramatically enhance the activity of the reaction to afford 3a with 89% yield and 82% e.e. (Table 2, entry 2 vs 1). Among all the additives examined (entries 3–6), 1,3-DCP was found to give the optimum results with desired product 3a obtained in 81% yield with 88% e.e. (entry 6). An increase in stereoselectivity was observed by fine-tuning the reaction conditions (entry 7).

### Substrate scope

Having optimized the reaction conditions for the regioisomeric allenylation, we next investigated the generality of the reaction substrates (Fig. 3A). A diverse array of propargylic carbonates 1 with various alkyl substituents performed well in the presence of nickel catalyst, affording the desired products in high yields and excellent

enantioselectivities (3a-3f). Propargylic carbonates with electron-donating or electron-withdrawing substituents on the benzene ring also proved to be suitable nucleophiles in the coupling reactions, thus furnishing the corresponding allenylphosphine oxides in 70–86% yield and 87–90% e.e. (3g-3j). The meta substituted propargylic carbonates also survived our catalytic conditions (3k and 3l). This method was compatible with furan and thiophene heterocycle-substituents (3 m and 3n). Moreover, various dialkyl substituted propargylic carbonates were well tolerated, leading to the corresponding products in good yields with excellent enantioselectivity (3o-3s).

The generality of the reaction with respect to the substituents of the phosphine oxides was also investigated (Fig. 3B). Both electron-donating and electron-withdrawing substituents were accommodated on the phenyl ring, and excellent levels of enantioselectivity were obtained (3t-3ab). In addition, the extension of the protocol to heterocyclic substituted phosphine oxide was also successful in affording the desired product in 67% yield and 87% e.e. (3ac).

Having established the stereoselective allenylation of phosphine oxides with (S,S)-(2-Naph)BDPP (L23) as the ligand, we further turned

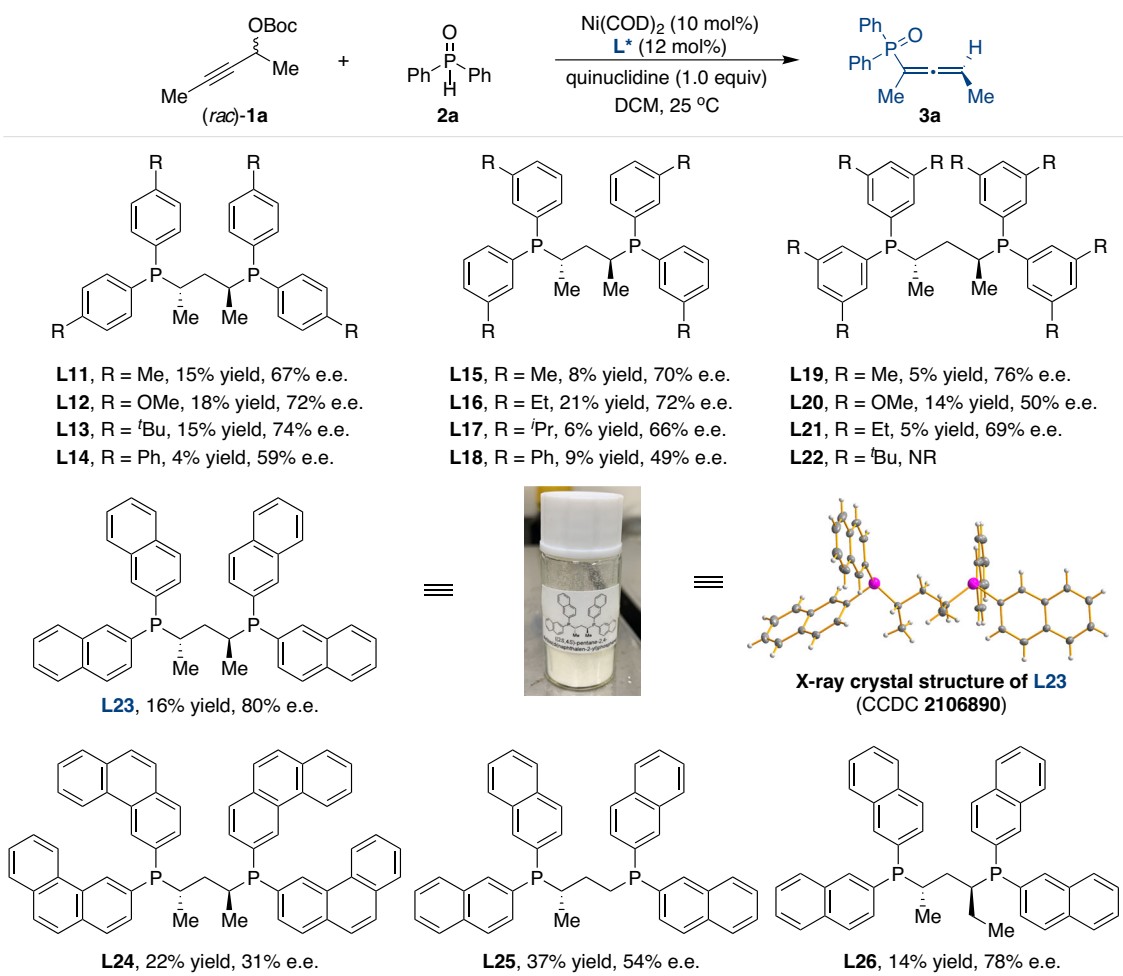

**Fig. 2 | Investigation of BDPP-type ligands in enantioselective allenylation.**
Reactions were conducted by using Ni(COD)$_2$ (10 mol%), **L*** (12 mol%), (*rac*)-**1a** (0.3 mmol), **2a** (0.1 mmol), and quinuclidine (1.0 equiv) in DCM (1 mL) at 25 °C for 24 h. e.e. values were determined by high-performance liquid chromatography analysis. The isolated yields of after chromatography are shown. NR = no reaction.

## Table 2 | Further optimization

| Entry | Additive | Yield (%) | e.e. of 3a (%) |
|---|---|---|---|
| 1 | — | 16 | 80 |
| 2 | HCO$_2$Li (0.2 equiv) | 89 | 82 |
| 3 | HCO$_2$Li (0.2 equiv) and MeOH (3.0 equiv) | 82 | 82 |
| 4 | HCO$_2$Li (0.2 equiv) and EtOH (3.0 equiv) | 84 | 82 |
| 5 | HCO$_2$Li (0.2 equiv) and TFE (3.0 equiv) | 71 | 87 |
| 6 | HCO$_2$Li (0.2 equiv) and 1,3-DCP (3.0 equiv) | 81 | 88 |
| 7[a] | HCO$_2$Li (0.2 equiv) and 1,3-DCP (3.0 equiv) | 90 | 90 |

Reactions were conducted by using Ni(COD)$_2$ (10 mol%), **L23** (12 mol%), (*rac*)-**1a** (0.3 mmol), **2a** (0.1 mmol), and quinuclidine (1.0 equiv) in DCM (1 mL) at 25 °C, 24 h. Isolated yield of after chromatography are shown. e.e. values were determined by high-performance liquid chromatography analysis (HPLC).
[a]Quinuclidine (3.0 equiv), 10 °C, 84 h. TFE = 2,2,2-thifluoroethanol. 1,3-DCP = 1,3-dichloropropan–2-ol.

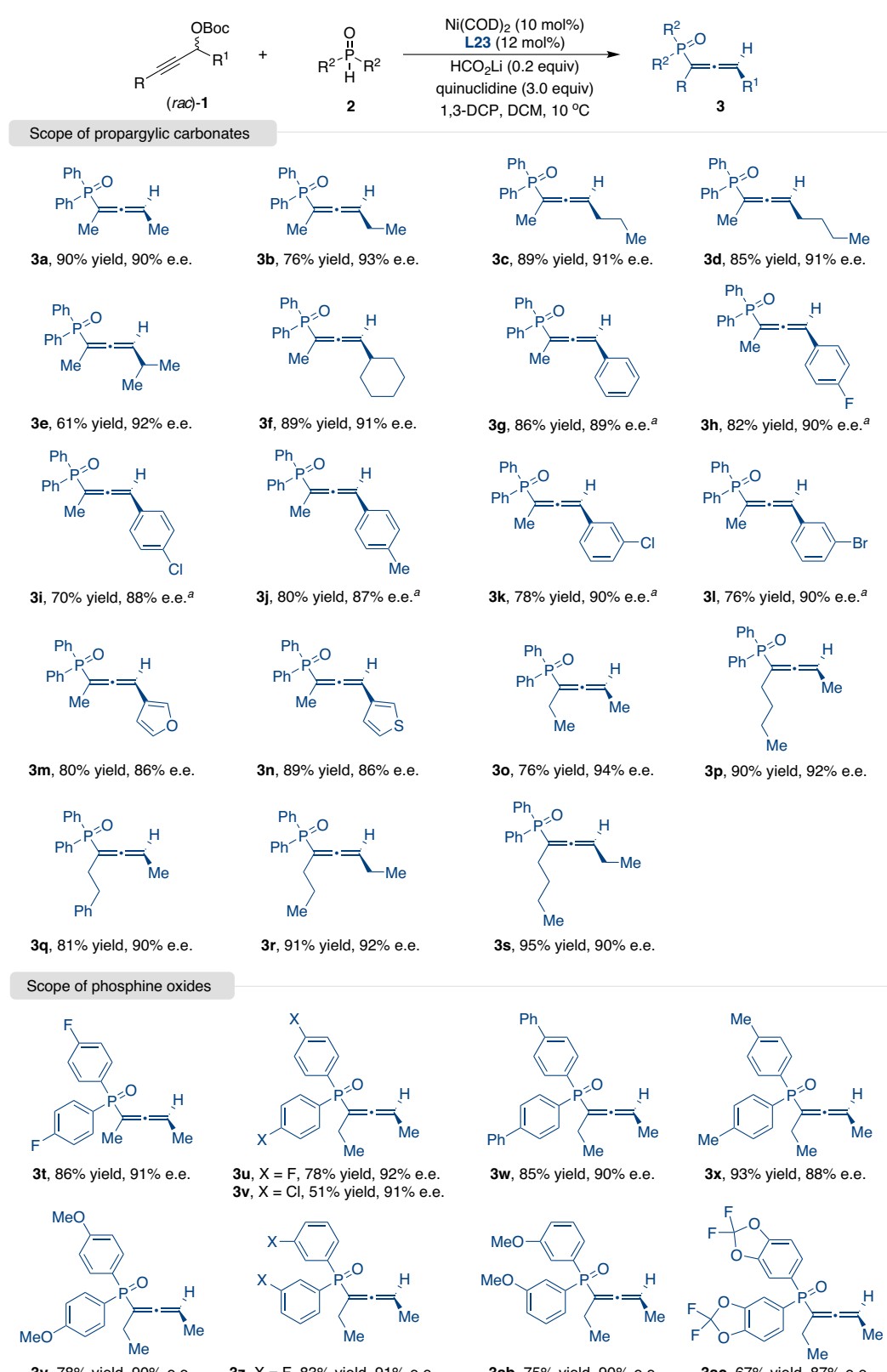

**Fig. 3 | Nickel-catalyzed enantioselective allenylation.** Reactions were performed by using Ni(COD)₂ (10 mol%), **L23** (12 mol%), (*rac*)-**1** (0.3 mmol), **2** (0.1 mmol), HCO₂Li (0.2 equiv), quinuclidine (3.0 equiv), and 1,3-DCP (30 μL) in DCM (1 mL) at 10 °C. Isolated yields after chromatography are shown. e.e. values were determined by HPLC analysis. *ᵃ*Quinuclidine (5.0 equiv), without 1,3-DCP, 0 °C.

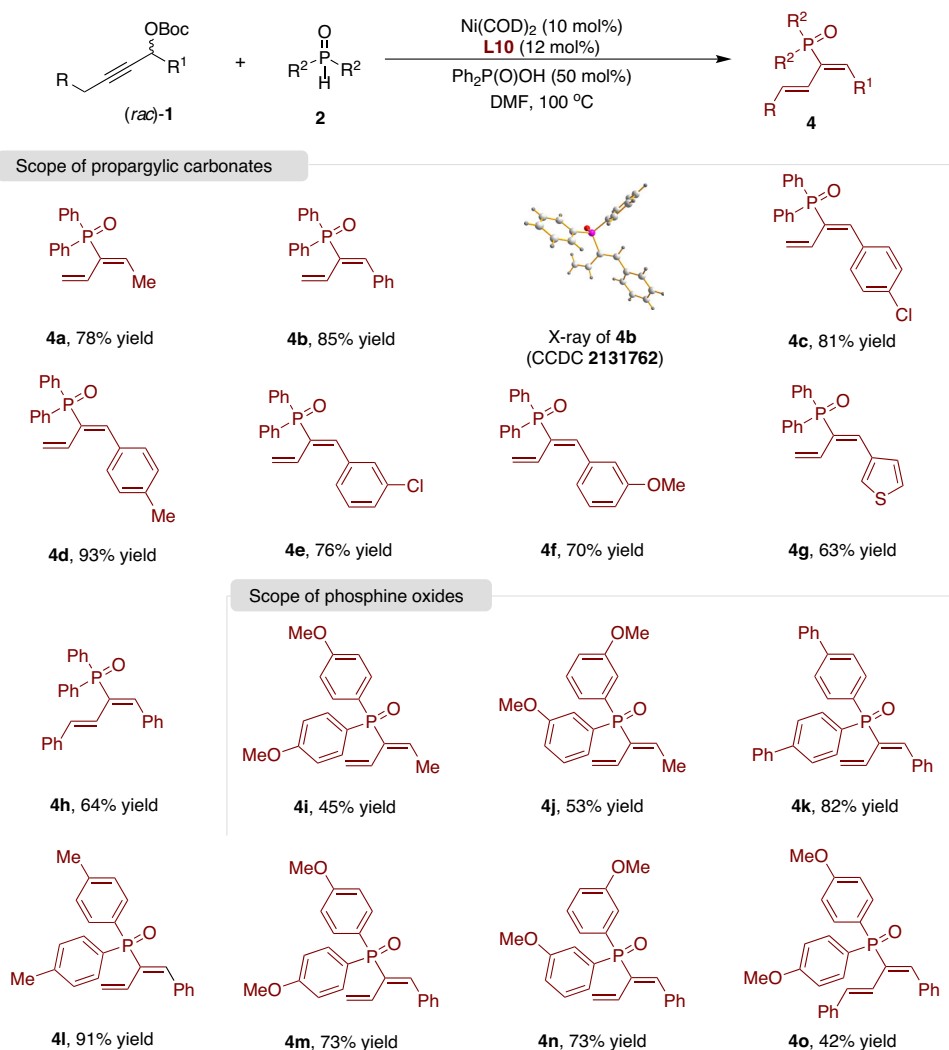

**Fig. 4 | Nickel-catalyzed 1,3-dienylation of phosphine oxides.** Reactions were conducted by using Ni(COD)₂ (10 mol%), **L10** (12 mol%), (*rac*)-**1** (0.12 mmol), **2** (0.1 mmol), and Ph₂P(O)OH (50 mol%) in DMF at 100 °C. Isolated yields after chromatography are shown.

our attention to complementary regioselective 1,3-dienylation with the phosphine ligand dcypbz (**L10**). Under the optimized reaction conditions (Table 1, entry 12), a broad range of propargylic carbonates and phosphine oxides were investigated (Fig. 4). Various substituted propargylic carbonates smoothly underwent this transformation, delivering phosphinoyl 1,3-butadienes **4** in high yields (**4a-4h**). The structure of **4b** was confirmed by X-ray single-crystal diffraction. Phosphine oxides with different substituents on the aromatic ring also afforded good results (**4i-4o**).

## Synthetic applications

To further demonstrate the practical utility of our protocol, a scale-up reaction was carried out under standard reaction conditions, furnishing **3a** in 71% yield with 90% e.e. (Fig. 5A). In addition, the treatment of allenylphosphine oxide **3a** with PhSeCl gave rise to desired selenohydroxylation product **5** in 86% yield with a maintained e.e.[64], and the absolute configuration of **5** was assigned by single-crystal X-ray diffraction analysis. Exposure of allenylphosphine oxide **3a** to iodine in MeCN/H₂O afforded **6**, which was subsequently treated with acetic anhydride to generate ester **7** in high yield without loss of enantiopurity. Subsequently, the Suzuki coupling of **7** with phenylboronic acid in MeCN afforded **8** with 90% e.e. in 86% yield. **7** could also be readily reduced and then oxidized in the presence of elemental selenium and sulfur to obtain the

corresponding phosphine selenide **10** and phosphine sulfide **11**, respectively. In addition, propargylic carbonate **1b** was subjected to 1,3-dienylation on a large scale, and the corresponding adduct **4b** was obtained in 65% yield (Fig. 5B). Hydrogenation of **4b** in the presence of a catalytic amount of Pd/C generated the corresponding phosphine oxide **12** in 87% yield. Furthermore, the treatment of **4b** with trichlorosilane furnished the corresponding phosphine **13**, which was then oxidized in the presence of elemental selenium to obtain phosphine selenide **14** in a high yield.

## Mechanistic studies

A series of experiments were conducted to gain a better understanding of the mechanistic details of this process (Fig. 6). The reactions carried out with the racemate and both enantiomers of propargylic carbonate **1p** produce good graphical overlay in the kinetic data (Fig. 6A), and under all conditions, (*S*)-**3p** was obtained in essentially identical e.e. (92% e.e.) monitored over different reaction times. Furthermore, reactions of (*rac*)−**1p**, (*R*)-**1p** (96% e.e.), and (*S*)-**1p** (93% e.e.) were separately conducted under the standard reaction conditions (Fig. 6B), and e.e. of the recovered **1p** was essentially unchanged throughout the reaction process, suggesting that the irreversible oxidative addition process occurred under the reaction conditions. All these results established that the chiral ligand effectively controls the absolute configuration of the

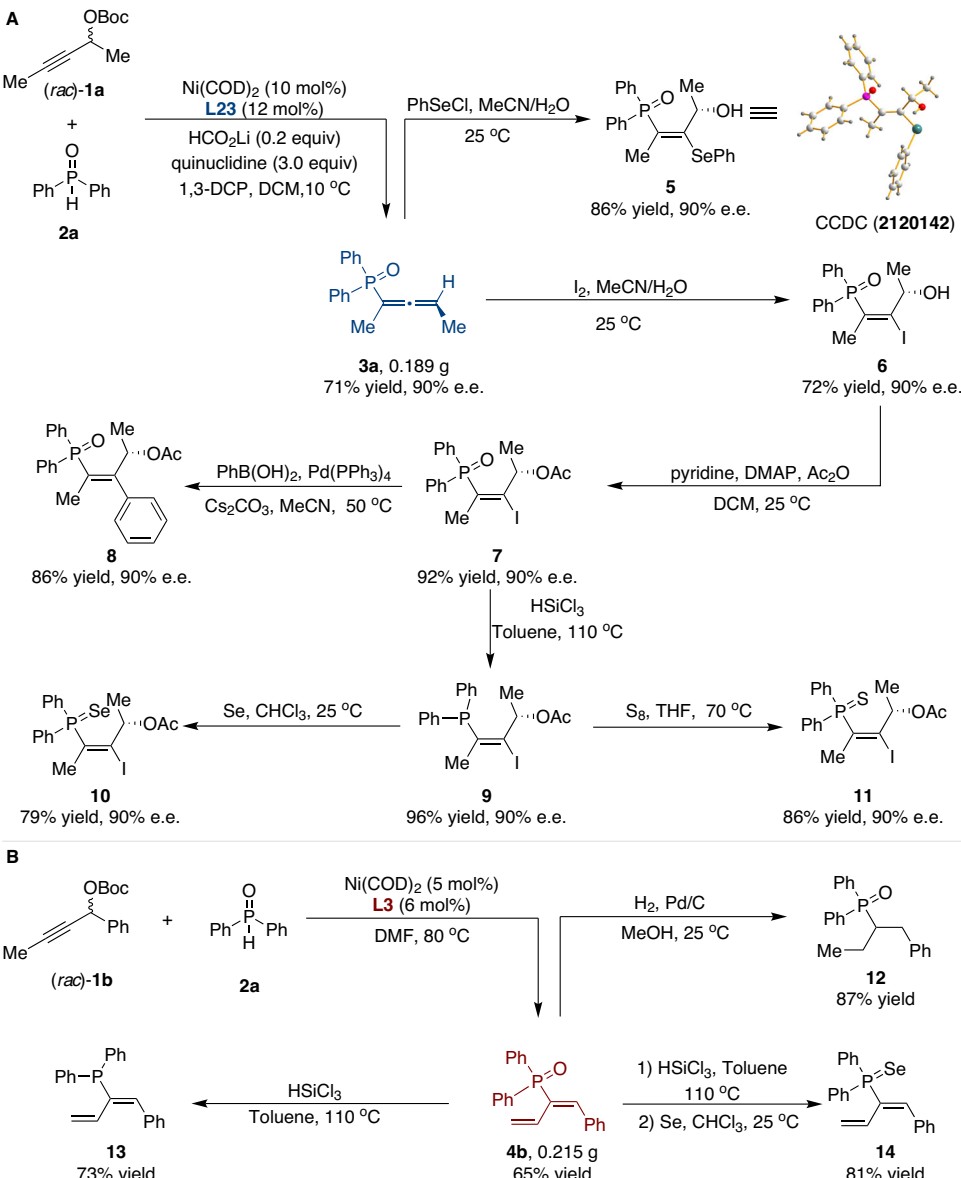

**Fig. 5 | Synthetic versatility of the catalytic system. A** Derivatization of product **3a**. **B** Derivatization of product **4b**.

product, regardless of the stereochemistry of the starting electrophilic substrates. The relationship between the e.e. value of **L23** and that of **3p** was then investigated using a ligand with different levels of enantiopurity (Fig. 6C, racemic, 20%, 40%, 50%, 60%, 80%, and >99% e.e.). Indeed, the nonlinear effect study revealed a linear relationship between the e.e. of **3p** and the enantiopurity of phosphine ligand **L23**, suggesting that one phosphine ligand is most likely involved in the active nickel species.

We achieved control over the reaction outcome in Ni-catalyzed reactions by slightly changing the reaction conditions. The selectivity between **3a** and **4a** could be controlled by the choice of the phosphine ligand (Table 1, entries 3 vs 8) and the acid additive (Table 1, entries 10 vs 12). These unusual results encouraged us to investigate the effect of acid or base effect on the regioselectivity (see Supplementary Information for details). As shown in Table 3, there were clear trends that allenylation pathway leading to **3a** with different phosphine ligands is favored with base additives (entries 1,4 vs 2,5), while the acid additive resulted in an improved formation of 1,3-dienylation product **4a** (entries 3,6 vs 2,5).

A proposed mechanism for the nickel-catalyzed coupling reactions of propargylic carbonates and phosphine oxides is illustrated in Fig. 7. The selectivity between the competing 1,3-dienylation and allenylation pathways was determined by the structure of the phosphine ligands and the additives[65–68]. As shown in Fig. 7, the proposed catalytic cycle for the generation of allenylphosphine oxide **3** begins with the Ni-mediated decarboxylation of propargylic carbonate **1**, and further delivers the allenylnickel complex **I** or **IV**[58–62]. Under basic conditions with **L23** as the ligand (left cycle), the in situ generated *tert*-butoxy anion could deprotonate hydroxydiphenylphosphane to generate the diphenylphosphinite anion. Therefore, the nucleophilicity of oxygen is higher than that of phosphine in **2′**, which might favor nickel-catalyzed *O*-propargylation[62]. Therefore, nucleophilic addition of hydroxyphosphine anion **2′** onto phosphine **L23**-bound allenylnickel intermediate **I** provides a stereoselective route to access chiral propargyl phosphinate **III**[62], and the following [2,3]-sigmatropic rearrangement furnishes the final allenylphosphine oxide **3**[69]. Alternatively, under acidic conditions with **L10** as the ligand (right

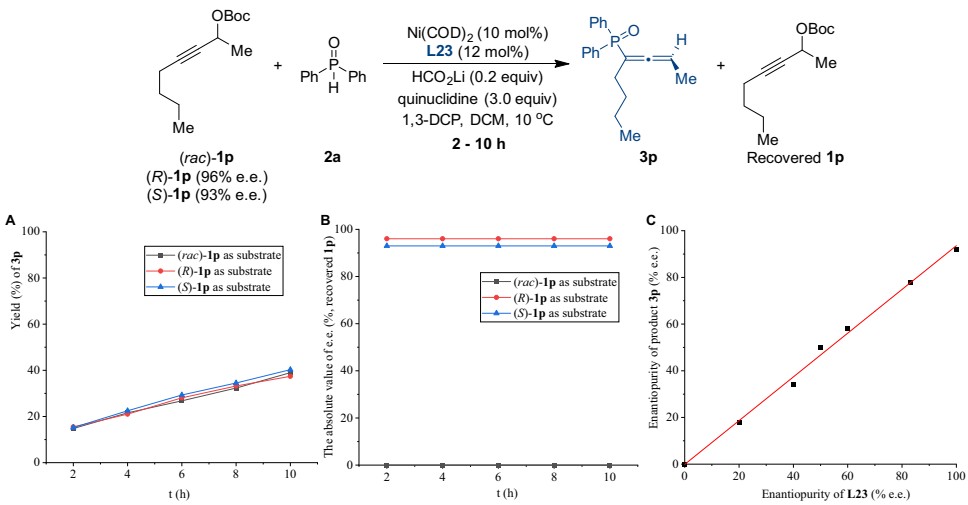

**Fig. 6 | Mechanistic studies. A** Time course of the kinetic resolution of **1p**. **B** e.e. of recovered **1p** at partial conversion. **C** Nonlinear relationship between the enantiopurity of phosphine ligand **L23** and adduct **3p**.

## Table 3 | Additive effect on the outcome of the reaction

| Entry | L | Additive | Yield (%) | 3a/4a |
|---|---|---|---|---|
| 1 | **L4** | quinuclidine | 95 | >20:1 |
| 2 | **L4** | none | 70 | 3:1 |
| 3 | **L4** | Ph$_2$P(O)OH | 39 | <1:20 |
| 4 | **L10** | quinuclidine | 93 | 4:1 |
| 5 | **L10** | none | 42 | 1:3 |
| 6 | **L10** | Ph$_2$P(O)OH | 40 | <1:20 |

Reactions were conducted by using Ni(COD)$_2$ (10 mol%), **L** (12 mol%), (*rac*)-**1a** (0.3 mmol), **2a** (0.1 mmol), in dioxane (1 mL) with quinuclidine (1.5 equiv) or diphenylphosphinic acid (0.5 equiv) as additives at 80 °C, 24 h. Isolated yield of the mixture **3a** and **4a** after chromatography are shown.

cycle), the nucleophilicity of phosphine is higher than that of oxygen in **2**"[57]. A possible diphenylphosphinic acid assisted proton shift process[70] is operated to generate an allyl nickel intermediate **V**[71], which is also consistent with the deuteration experiments (see Supplementary Information for details). The ligand exchange of intermediate **V** with **2"** and subsequent reductive elimination of nickel complex **VI**[71] regenerates the nickel catalyst and furnishes final 1,3-dienyl isomer **4**.

## Discussion

In summary, a nickel-catalyzed 1,3-dienylation and enantioselective allenylation of phosphine oxides have been developed. The employment of different phosphine ligands allows for the highly regio- and stereoselective formation of allenylphosphine oxides and phosphinoyl 1,3-butadienes from identical starting materials. Furthermore, this reaction represents the successful example of catalytic enantioselective construction of allenylphosphoryl frameworks with excellent enantioselectivity (up to 94% e.e.) using a highly efficient type of chiral BDPP-type ligand. The high efficiency, stereoselectivity, and operational simplicity of this transformation, coupled with the rational design of chiral ligands, are expected to render this method a valuable tool in asymmetric synthesis.

## Methods

### General procedure for nickel-catalyzed enantioselective allenylation

In a 10 mL Schlenk tube, Ni(COD)$_2$ (2.8 mg, 0.01 mmol, 10 mol%), **L23** (7.6 mg, 0.012 mmol, 12 mol%), and 1,3-DCP (30 μL) were stirred in 1 mL anhydrous DCM under argon at room temperature for 20 min. Propargylic carbonate **1** (0.3 mmol), phosphine oxide **2** (0.1 mmol, 1.0 equiv), quinuclidine (33.3 mg, 0.3 mmol), and HCO$_2$Li (1.4 mg, 0.02 mmol) were then added successively. The reaction mixture was stirred at 10 °C until the reaction was complete (monitored by TLC). The reaction mixture was concentrated, and the residue was purified by column chromatography to afford the corresponding product **3**.

### General procedure for nickel-catalyzed 1,3-dienylation

In a 10 mL Schlenk tube, Ni(COD)$_2$ (2.8 mg, 0.01 mmol, 10 mol%) and dcypbz **L10** (5.6 mg, 0.012 mmol, 12 mol%) were stirred in 1 mL anhydrous DMF under argon at 80 °C for 10 min. Propargylic carbonate **1** (0.12 mmol), phosphine oxide **2** (0.1 mmol, 1.0 equiv) and HOP(O)Ph$_2$ (10.9 mg, 0.05 mmol) were then added successively. The reaction mixture was stirred at 100 °C until the reaction was complete (monitored by TLC). The reaction mixture was concentrated, and the residue was purified by flash column chromatography on silica gel to give the desired product **4**.

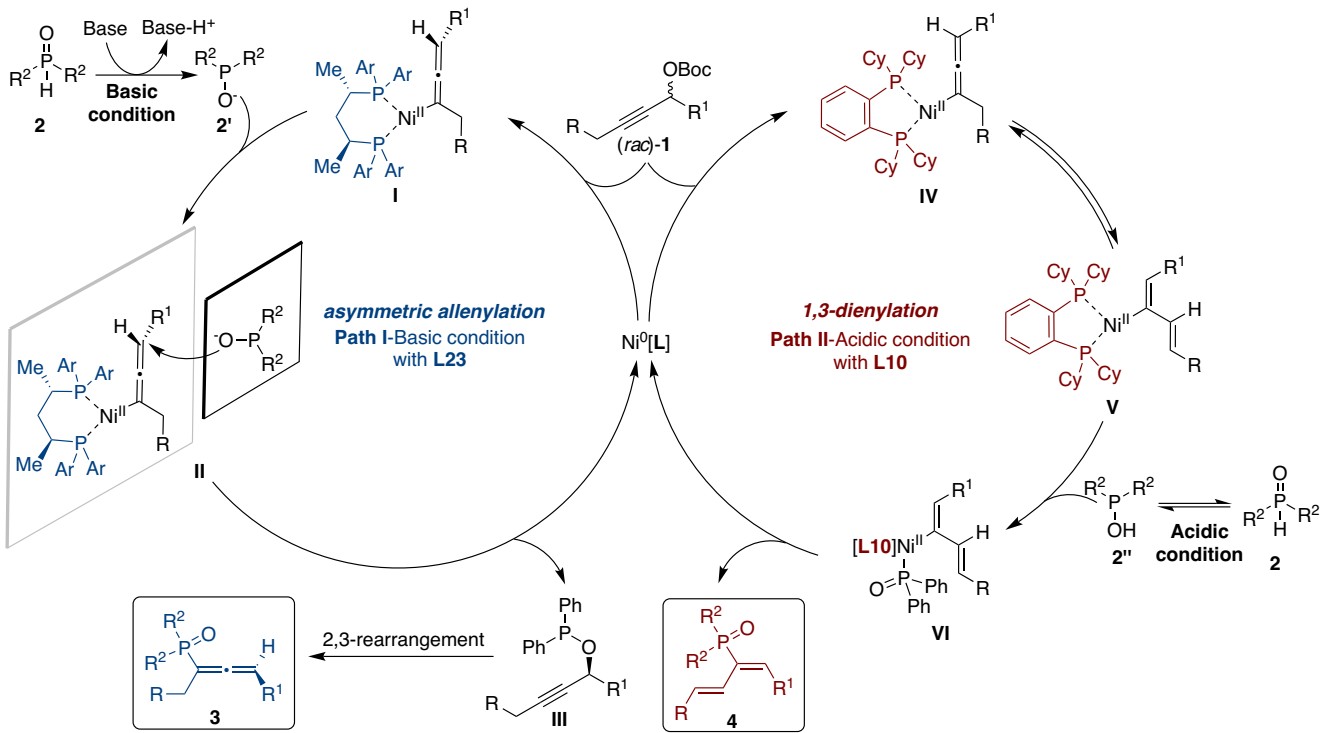

**Fig. 7 | Proposed catalytic cycle.** Pathways are shown for the catalytically afforded reactive species from propargylic carbonate with nickel catalysts. Under basic condition, the phosphine **L23**-bound allenylnickel intermediate **I** can react with hydroxyphosphine anion **2′** to produce **3** (left cycle). Under the acidic condition, an allyl nickel intermediate **V** undergoes the reaction with **2″** to furnishes the final 1,3-dienyl isomer **4** (right cycle).

## Data availability

All data generated or analyzed during this study are included in the published Article and Supplementary Information. Crystallographic data for the structures reported in this Article have been deposited at the Cambridge Crystallographic Data Centre, under deposition numbers CCDC 2106890 (**L23**), 2131762 (**4b**) and 2120142 (**5**). Copies of the data can be obtained free of charge via https://www.ccdc.cam.ac.uk/structures/.

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

## Acknowledgements

The authors acknowledge financial support from the National Natural Science Foundation of China (grant no. 21971227, C.G.) and the Fundamental Research Funds for the Central Universities (WK2340000090, C.G.). Prof. Yu Lan of Zhengzhou University is acknowledged for valuable discussions.

## Author contributions

C.G. conceived and designed the study, and wrote the paper. J.Z., X.C., X.X., H.W., and L.P. performed the experiments and analyzed the data. All authors discussed the results and commented on the manuscript.

## Competing interests

The authors declare no competing interests.
