## [Peer Review File · Nature Communications]

REVIEWER COMMENTS

Reviewer #1 (Remarks to the Author):

This paper reported a chiral ligand-controlled nickel-catalyzed 1,3-dienylation and enantioselective allenylation of phosphine oxides. This looks very interesting, however the present result is fundamental and needs to make further studies before publication.

1. Using the expensive chiral Duphos as ligand to prompt the achiral reaction of 1,3-dienylation is wasteful. Author should screen more achiral ligands and obtain more useful results. Otherwise, the author ought to delete the part of the 1,3-dienylation in the manuscript.

2. The novelty and key scientific about this paper is ligand-control. How does the ligand control this regioselectivity? The author should make more control experiments such as deuterium reaction, mass spectra and DFT calculations to testify the proposed mechanism.

3. Does the heterocycle of propargylic carbonates suit for this reaction?

4. Author should provide more examples of derivatization to display the usefulness of the product.

Reviewer #2 (Remarks to the Author):

This manuscript describes the nickel-catalyzed reactions of propargyl alcohol derivatives with diarylphosphine oxides, which can lead to phosphorus-oxide substituted chiral allenes or dienes by using different types of ligands. The construction of chiral allenes from propargyl alcohol derivatives has been reported by Prof. Ma's groups recently (ref. 45-50). The racemic version of this reaction has been well achieved using metals such as palladium, copper and silver (ref. 52-56). The authors have previously achieved asymmetric propargylic substitution reactions using nickel catalysts with various nucleophilic reagents (ref. 60-62), and the difference in this paper is that when phosphorus oxides were used as the nucleophilic reagent, a chiral allene was obtained. Also, it should be noted that the nickel catalysts combined with chiral BDPP or Duphos ligands have been reported in the asymmetric allylic and benzyl substitution reactions of phosphorus oxides (Zhang et al. JACS, 2019, 141, 16584; OL, 2022, 24, 1258). Therefore, the major concept is not truly new. I do not recommend publication of this work in "Nature Communication".

Reviewer #3 (Remarks to the Author):

In the manuscript, the authors described a nickel-catalyzed ligand-controlled chemo- and regioselective allenylation and 1,3-dienylation of phosphine oxides with propargylic carbonates. The results are highly interesting. By the development of new BDPP-type diphosphine ligands, the authors realized the nickel-catalyzed asymmetric allenylation which led to allenylphosphine oxides as the only product in good to high enantioselectivity. The use of DuPhos instead of BDPP-type ligands, a complete switch in selectivity from allenylphosphine oxides to 1,3-dienyl isomers was observed and realized. The reaction displayed broad substrate scope, and a reasonable mechanism has been proposed to rationalize the observed selectivity. Furthermore, the manuscript is well organized. This reviewer therefore recommends it for the publication in Nat. Commun. after a minor revision.

1. The optimized condition for allenylation should be use of 1,3-DCP as that in Fig 3 and the comment for Table 2, while the best result in Table 2 was obtained with MeOH (entry 7)?
2. A model for the observed stereochemistry of allenylation is suggested.

Point-by-point response

for

Nickel-Catalyzed Switchable 1,3-Dienylation and Enantioselective Allenylation of Phosphine Oxides

Jiayin Zhang, Xihao Chang, Xianghong Xu, Hongyi Wang, Lingzi Peng, Chang Guo*

Manuscript number: NCOMMS-22-01139

Reply to comments by Reviewer 1

1. This paper reported a chiral ligand-controlled nickel-catalyzed 1,3-dienylation and enantioselective allenylation of phosphine oxides. This looks very interesting, however the present result is fundamental and needs to make further studies before publication.

Answer: We appreciate reviewer 1 for the favorable comments and helpful suggestions! We studied these comments carefully and made corrections as following. Revised portions are highlighted in our revised manuscript.

2. Using the expensive chiral Duphos as ligand to prompt the achiral reaction of 1,3-dienylation

is wasteful. Author should screened more achiral ligands and obtained more useful result. Otherwise author ought to delete the part of the 1,3-dienylation in the manuscript.

Answer: As suggested by Reviewer 1, a variety of achiral phosphine ligands (**SL1-SL4**, **L9**, and **L10**) was evaluated to prompt the reaction of 1,3-dienylation. To our great delight, we found that the desired 1,3-dienylation product **4a** could be obtained in good yield and good regioselectivity using **L10** as ligand along with 50 mol% of diphenylphosphinic acid as the acid additive (Table S1, entry 7). We have included these results in our revised manuscript (Table 1) and Supplementary Information (Page S5, Table S1).

Table S1. Optimization for the 1,3-dienylation.

Entry	L	solvent	Additive (0.5 equiv)	T (°C)	3a/4a	yield (%)
1	SL-1	dioxane	-	80	-	NR
2	SL-2	dioxane	-	80	4:1	32
3	SL-3	dioxane	-	80	-	NR
4	SL-4	dioxane	-	80	-	NR
5	L9	dioxane	-	80	3:1	31
6	L10	dioxane	-	80	1:3	42
7	L10	DMF	Ph ₂ P(O)OH	100	<1:20	78

Reactions were conducted by using Ni(COD)₂ (10 mol%), **L** (12 mol%), (*rac*)-**1a** (0.3 mmol), and **2a** (0.1 mmol) for 24 h. Isolated yield of the mixture **3a** and **4a** after chromatography are shown.

We next investigated the generality of the reaction scope with the phosphine ligand dcy pbz (**L10**). Under the optimized reaction conditions (Table 1, entries 11 and 12), a broad range of propargylic carbonates and phosphine oxides were investigated with the phosphine ligand dcy pbz (**L10**). Various substituted propargylic carbonates smoothly underwent this transformation, delivering phosphinoyl 1,3-butadienes **4** in high yields (**4a-4h**). We have included these results in our revised manuscript (Fig. 4).

Fig. 4 | Nickel-catalyzed 1,3-dienylation of phosphine oxides.

3. The novelty and key scientific about this paper is ligand-control. How does the ligand controlled this regioselectivity? the author should make more control experiments such as deuterium reaction, mass spectra and DFT calculations to testify the proposed mechanism.

Answer: We appreciate reviewer 1 for the favorable comments and the questions raised regarding the mechanism were really helpful. As suggested by Reviewer 1, we have started a thorough investigation of the underlying mechanism (mass spectra, acid and base effect, deuterium reaction, control experiments, and DFT calculations, see Supplementary section 2 for details).

ESI-MS studies:

In a 10 mL Schlenk tube, Ni(COD)₂ (2.8 mg, 0.01 mmol, 10 mol%) and dcybpz **L10** (5.6 mg,

0.012 mmol, 12 mol%) were stirred in 1 mL anhydrous DMF under argon at 50 °C for 20 min. Tert-butyl (1-phenylbut-2-yn-1-yl) carbonate were then added successively. The reaction mixture was stirred at 25 °C for 30 min and the solution was determined by ESI-MS.

The proposed allenynickel intermediate in the crude reaction solution was supported by ESI-MS analysis for our standard reaction. The signals at $m/z = 657.3283$ with its characteristic isotope distribution by mass spectrometry matched with the calculated patterns for the intermediate $[\text{Ni}(\mathbf{L10})(\text{allenyl})]^+$. We have included these results in our revised Supplementary Information (Pages S14 and S15).

ESI-MS:

$[\text{C}_{40}\text{H}_{57}\text{NiP}_2]^+$

Calcd : 657.3283

Found : 657.3290

Fit the isotope distribution:

Acid and base effect:

Allenylation pathway leading to **3a** is favored with base additive (Table S2, entries 1,4 vs 2,5), while the acid additive resulted in an improved formation of 1,3-dienylation product **4a** (Table S2, entries 3,6 vs 2,5). To further support our hypothesis, we screened different phosphine ligands in the reaction between **1a** and **2a** with different additives in order to investigate the relationship between the acid and base effect to the product distribution. Using **L3** as the phosphine ligand, we found the ratio of the allenylation product **3a** to the 1,3-dienylation adduct **4a** decreased dramatically to 1:1 (entries 7 vs 8) when the base additive was used. In contrast, with **L8** as the ligand, the acid additive gave an increased amount of the 1,3-dienylation adduct **4a** (entries 9 vs 10). These results indicate that the acid additive results in an increased amount of **4a**, and the base additive favors the allenylation product **3a**. We have included these results in our revised manuscript (Table 3) and Supplementary Information (Table S2).

Table S2 | Additive effect on the outcome of the reaction.

Entry	L	Additive	Yield (%)	3a/4a
1	L4	quinuclidine	95	>20:1
2	L4	none	70	3:1
3	L4	Ph ₂ P(O)OH	39	<1:20
4	L10	quinuclidine	93	4:1
5	L10	none	42	1:3
6	L10	Ph ₂ P(O)OH	40	<1:20
7	L3	none	45	<1:20
8	L3	quinuclidine	89	1:1
9	L8	none	52	>20:1
10	L8	Ph ₂ P(O)OH	50	3:1

Reactions were conducted by using Ni(COD)₂ (10 mol%), L (12 mol%), (*rac*)-**1a** (0.3 mmol), **2a** (0.1 mmol), and quinuclidine (1.5 equiv) or diphenylphosphinic acid (0.5 equiv) in dioxane (1 mL) at 80 °C, 24 h. Isolated yield of the mixture **3a** and **4a** after chromatography are shown.

Deuterium-labeling experiments:

General procedure for deuterium labeling experiments of the allenylation: In a 10 mL Schlenk tube, Ni(COD)₂ (2.8 mg, 0.01 mmol, 10 mol%) and **L23** (7.6 mg, 0.012 mmol, 12 mol%) were stirred in anhydrous DCM (1 mL) under argon at room temperature for 20 min. Propargylic carbonate **1b** (0.3 mmol), phosphine oxide **2** (0.1 mmol, 1.0 equiv), quinuclidine (33.3 mg, 0.3 mmol), HCO₂Li (1.4 mg, 0.02 mmol) and methanol-*d* (1.0 mmol) were then added successively. The reaction mixture was stirred at 25 °C until the reaction was complete (monitored by TLC). The mixture was subjected to silica gel column chromatography directly for purification.

General procedure for deuterium labeling experiments of the 1,3-dienylation: In a 10 mL Schlenk tube, Ni(COD)₂ (1.4 mg, 0.005 mmol, 5 mol%) and ⁱPrDuphos **L3** (2.5 mg, 0.006 mmol, 6 mol%) were stirred in anhydrous DMF (1 mL) under argon at 80 °C for 10 min. Propargylic carbonate **1b** (0.12 mmol), phosphine oxide **2** (0.1 mmol, 1.0 equiv) and methanol-*d* (1.0 mmol) were then added successively. The reaction mixture was stirred at 80 °C until the reaction was complete (monitored by TLC). The reaction mixture was concentrated, and the residue was purified by flash column chromatography on silica gel to give the desired product.

Initially, deuteration experiments for the nickel-catalyzed allenylation were conducted (Fig. S6a and Fig. 6b). Propargylic carbonate **1b** was subjected to the nickel-catalyzed allenylation system in MeOD. After the reaction, **3g** was obtained as the major stereoisomer, and deuteration of the product **3g-D** was not observed (Fig. S6a). In addition, the nickel-catalyzed allenylation of propargylic carbonate **1b-D** (Figure 6b), which bears the deuterium atom at the propargylic position, provided the product **3g-D** (0.96 D). These results indicate that methanol is not involved in the reaction mechanism of the allenylation reaction.

In addition, deuteration experiments for the nickel-catalyzed 1,3-dienylation were also conducted. The treatment of **1b** with MeOD under the nickel-catalyzed 1,3-dienylation system gave the deuterium-incorporated diene **4b-D₃** a high D/H ratio (Fig. S6c, 0.7 D; Fig. S6d, 0.63 D), which is also consistent with the proposed mechanism in the acid condition (Fig. 7). In addition, the propargylic carbonate **1b-D** was subject to the 1,3-dienylation reaction, leading to the product **4b-D** with similar deuterium incorporation (Fig. S6e), which suggested that hydrogen at the propargylic position did not contribute to the protonation event. In addition, we could not observe deuterium-incorporated products when the control experiment using **4b** and MeOD was conducted (Fig. S6f).

To explain the deuterium incorporation of the products (Fig. S6d), we propose that a possible

diphenylphosphinic acid-assisted proton shift process is operated reversibly to generate an allyl nickel intermediate.

Fig. S6. Deuteration experiments for the nickel-catalyzed 1,3-dienylation and allenylation of phosphine oxides.

^1H NMR analyses **4b-D₃** were shown in Scheme S1 (page S12 and Page 13). We have included these results in our revised Supplementary Information (Fig. S6).

¹H NMR of (rac)-1b-D

¹H NMR of 3g-D

¹H NMR of 4b-D₃

¹H NMR of 4b-D

DFT calculations:

We achieved control over the reaction outcome in nickel-catalyzed reactions by slight

changing the reaction conditions, and the selectivity between 3 and 4 could be controlled by the choice of acid or base additive. To understand the reaction mechanism of acid and base effect, DFT calculations were also conducted. Initially, the nickel complex activates the propargylic carbonates to generate the allenynickel species. In the base condition, a strong base, *tert*-butoxy anion would be generated *in situ* in the catalytic cycle with propargylic carbonate as substrate, which can deprotonate hydroxydiphenylphosphane to generate the diphenylphosphinite anion (*Inorg. Chem.* **2012**, *51*, 7903). However, in the presence of diphenylphosphinic acid, a possible diphenylphosphinic acid-assisted proton shift process (*J. Am. Chem. Soc.* **2007**, *129*, 3470) is operated to generate an allyl nickel intermediate (*Chem. Sci.* **2022**, *13*, 4095), which is also consistent with the deuteration experiments (Fig. S6d). The ligand exchange of intermediate **V** with **2''** and subsequent reductive elimination of the nickel complex regenerates the nickel catalyst and furnishes the final 1,3-dienyl isomer **4**.

To gain insight into the selective reaction mechanism, the nucleophilicity of **I** and **II** was calculated. We found that the nucleophilicity of oxygen is higher than that of phosphine in the base condition. In contrast, the nucleophilicity of phosphine is much higher than that of oxygen in the acid condition. Recently, we developed the nickel-catalyzed *O*-propargylation to afford an array of enantioenriched *O*-propargyl hydroxylamines (*J. Am. Chem. Soc.* **2021**, *143*, 21048), which suggested that *O*-nucleophiles might favor the propargylic substitution reaction. Alternative, under acidic conditions, hydroxyphosphine **2''** attacks the central carbon of the π -allenyl moiety of nickel complex, which was protonated by diphenylphosphinic acid to furnish the π -allylnickel intermediate (*J. Am. Chem. Soc.* **2019**, *141*, 84).

To gain more information about the possible intermediates for the allenylation of phosphine oxides, we turned our attention to studying the [2,3]-sigmatropic rearrangement. The enantioenriched propargylic alcohol (*S*-**S1**, 95% e.e.) was selectively converted into the *S*-**3p** in good yield with 95% e.e., indicating the efficient transformation of allenylphosphine oxide from the chiral propargyl phosphinate via the [2,3]-sigmatropic rearrangement (Page S15), which is also consistent with the results in the base condition and propargyl phosphinate might be initially generated.

In addition, deuteration experiments suggested that a possible diphenylphosphinic acid assisted proton shift process is operated reversibly to generate an allenyl nickel intermediate to furnish the final 1,3-dienyl isomer **4**.

Proposed mechanism:

A reasonable mechanism for the nickel-catalyzed switchable coupling reactions of propargylic carbonates and phosphine oxides is illustrated in Fig. 7. As shown in Fig. 7 in our revised manuscript, the proposed catalytic cycle for the generation of allenylphosphine oxide **3** begins with the Ni-mediated decarboxylation of propargylic carbonate **1**, and further delivers the allenylnickel complex **I** or **IV**. The nucleophilicity of oxygen is higher than that of phosphine in the base condition, while the nucleophilicity of phosphine is much higher than that of oxygen in the acid condition. Therefore,

under basic conditions with (left cycle), the *in situ* generated *tert*-butoxy anion could deprotonate hydroxydiphenylphosphane to generate the diphenylphosphinite anion, which might favor the nickel-catalyzed *O*-propargylation. Therefore, nucleophilic addition of the hydroxyphosphine anion **2'** onto the phosphine **L23**-bound allenylnickel intermediate **I** provide a stereoselective route to access chiral propargyl phosphinate **III**, and the following [2,3]-sigmatropic rearrangement furnishes the final allenylphosphine oxide **3**. Alternatively, under the acidic condition with **L10** as the ligand (right cycle), a possible diphenylphosphinic acid-assisted proton shift process is operated to generate an allyl nickel intermediate **V**, which is also consistent with the deuteration experiments (Fig. S6d). The ligand exchange of intermediate **V** with **2''** and subsequent reductive elimination of the nickel complex **VI** regenerates the nickel catalyst and furnishes the final 1,3-dienyl isomer **4**.

In conclusion, the acid and base additive effect the nucleophilic sites of phosphine oxide **2**, and the phosphine ligand plays a vital role in controlling the reaction pathway (allenylnickel complex vs allyl nickel intermediate). Both key factors determine the selectivity of the products. Therefore, the selectivity between the competing 1,3-dienylation and allenylation pathways was determined by the structure of the phosphine ligands and the acid or base additives.

4. Does the heterocycle of propargylic carbonates suitable for this reaction?

Answer: As suggested by Reviewer 1, we tested the suitability of substrates with heterocycle of propargylic carbonate for the reaction. Remarkably, this method was compatible with furan and thiophene heterocycle-substituents, affording the desired products in high yields and good enantioselectivities (Fig. 3, **3m** and **3n**). In addition, heterocyclic substituted propargylic carbonate smoothly underwent the 1,3-dienylation transformation, delivering phosphinoyl 1,3-butadienes **4g** with the phosphine ligand dcy pbz (**L10**, 63% yield) in high yield (Fig. 4). We have included these results in our revised manuscript.

5. Author should provides more examples of derivatization to display the usefulness of the product.

Answer: As suggested by Reviewer 1, we offer more examples of derivatization to demonstrate the utility of the current protocol. The treatment of allenylphosphine oxide **3a** with PhSeCl gave rise to desired selenohydroxylation product **5** in 86% yield with maintained e.e., and the absolute configuration of **5** was assigned by single-crystal X-ray diffraction analysis. Exposure of allenylphosphine oxide **3a** to iodine in MeCN/H₂O afforded **6**, which was subsequently treated with acetic anhydride to generate ester **7** in high yield without loss of enantiopurity. Subsequently, the Suzuki coupling of **7** with phenylboronic acid in MeCN afforded **8** with 90% e.e. in 92% yield. **7** could also be readily reduced, and then oxidized in the presence of elemental selenium and sulfur to obtain the corresponding phosphine selenide **10** and phosphine sulfide **11**, respectively. In addition, propargylic carbonate **1b** was subjected to the 1,3-dienylation on a large scale, and the corresponding adduct **4b** was obtained with 65% yield (Fig. 5B). Hydrogenation of **4b** in the presence of a catalytic amount of Pd/C generated the corresponding phosphine oxide **12** in 87% yield. Furthermore, the treatment of **4b** with trichlorosilane furnished the corresponding phosphine **13**, which was then oxidized in the presence of elemental selenium to obtain the phosphine selenide **14** in a high yield. We have included these results in our revised manuscript (Fig. 5).

Fig. 5 | Synthetic versatility of the catalytic system. A. Derivatization of the product 3a. B. Derivatization of the product 4b.

Reply to comments by Reviewer 2

- This manuscript describes the nickel-catalysed reactions of propargyl alcohol derivatives with diarylphosphine oxides, which can lead to phosphorus-oxide substituted chiral allenes or dienes by using different types of ligands. The construction of chiral allenes from propargyl alcohol derivatives has been reported by Prof. Ma's groups recently (ref. 45-50). The racemic version of this reaction has been well achieved using metals such as palladium, copper and silver (ref. 52-56).

Answer: We appreciate Reviewer 2 for comments. We have made all the necessary amendments as suggested in our revised manuscript and revised supporting information. Prof. Ma's group reported elegant Pd-catalyzed transformations to generate chiral allenes via reductive elimination (ref. 45-50). As mentioned by Reviewer 2, only a racemic version of this reaction has been achieved.

Therefore, we designed and synthesized a series of chiral phosphine ligands to meet the aforementioned challenges. More importantly, according to the mechanistic studies and control experiments of our current project, a substitution of hydroxyphosphine with allenylnickel intermediate affords the chiral propargyl phosphinate, and the following [2,3]-sigmatropic rearrangement furnishes the final allenylphosphine oxide. In addition, a switchable nickel catalyzed 1,3-dienylation and enantioselective allenylation of phosphine oxides was developed. Moreover, a comprehensive investigation suggested that the allenylation pathway leading to **3** is favored with base additive (Table 3, entries 1,4 vs 2,5), while the acid additive resulted in an improved formation of 1,3-dienylation product **4a**. In conclusion, the acid and base additive effect the nucleophilic sites of phosphine oxide **2**, and the phosphine ligand plays a vital role in controlling the reaction pathway (allenylnickel complex vs allyl nickel intermediate). Both key factors determine the selectivity of the products. Therefore, the selectivity between the competing 1,3-dienylation and allenylation pathways was determined by the structure of the phosphine ligands and the acid or base additives.

2. The authors have previously achieved asymmetric propargylic substitution reactions using nickel catalysts with various nucleophilic reagents (ref. 60-62), and the difference in this paper is that when phosphorus oxides were used as the nucleophilic reagent, a chiral allene was obtained.

Answer: As mentioned by Reviewer 2, the Kawatsura group and our group reported the nickel-catalyzed propargylic substitution reactions. However, only chiral propargylic products could be generated and the scope of nucleophiles that can efficiently participate in the reaction remains narrow. Therefore, we speculated that the nickel-catalytic system with the rational design and development of new chiral ligands would meet the challenges above to achieve carbon-phosphorus bond formation in an asymmetric manner. In addition, the employment of different phosphine ligands allows for the highly regio- and stereoselective formation of allenylphosphine oxides and phosphinoyl 1,3-butadienes from identical starting materials. The high efficiency, stereoselectivity, and operational simplicity of this transformation, coupled with the rational design of novel chiral ligands, are expected to render this method a valuable tool in asymmetric synthesis. In addition, we offer more examples of derivatization to demonstrate the utility of the current protocol (Fig. 5).

3. Also, it should be noted that the nickel catalysts combined with chiral BDPP or Duphos ligands have been reported in the asymmetric allylic and benzyl substitution reactions of phosphorus oxides (Zhang et al. JACS, 2019, 141, 16584; OL, 2022, 24, 1258).

Answer: We appreciate reviewer 2 for the favorable comments. The two literatures focused on the P-stereogenic tertiary phosphine oxides. Our protocol shows the catalytic enantioselective construction of allenylphoryl frameworks with axial chirality. In addition, a switchable nickel catalyzed 1,3-dienylation and enantioselective allenylation of phosphine oxides was developed. We also found that the acid additive results in an increased amount of 1,3-dienylation product **4**, and the base additive favors the allenylation product **3**. In addition, the acid and base additive effect the nucleophilic sites of phosphine oxide **2**, and the phosphine ligand plays a vital role in controlling the reaction pathway (allenylnickel complex vs allyl nickel intermediate). Therefore, the selectivity between the competing 1,3-dienylation and allenylation pathways was determined by the structure of the phosphine ligands and the acid or base additives. As suggested by Reviewer 2, we have cited related literature in our revised manuscript (References 66 and 67).

Reply to comments by Reviewer 3

1. In the manuscript, the authors described a nickel-catalyzed ligand-controlled chemo- and regioselective allenylation and 1,3-dienylation of phosphine oxides with propargylic carbonates. The results are highly interesting. By the development of new BDPP-type diphosphine ligands, the authors realized the nickel-catalyzed asymmetric allenylation which led to allenylphosphine oxides as the only product in good to high enantioselectivity. The use of DuPhos instead of BDPP-type ligands, a complete switch in selectivity from allenylphosphine oxides to 1,3-dienyl isomers was observed and realized. The reaction displayed broad substrate scope, and a reasonable mechanism has been proposed to rationalize the observed selectivity. Furthermore, the manuscript is well organized. This reviewer therefore recommends it for the publication in Nat. Commun. after a minor revision.

Answer: We appreciate reviewer 3 for the favorable comments and helpful suggestions! These comments are greatly valuable and helpful for revising and improving our paper. We have made all the necessary amendments as suggested in our revised manuscript and revised Supplementary Information.

2. The optimized condition for allenylation should be use of 1,3-DCP as that in Fig 3 and the comment for Table 2, while the best result in Table 2 was obtained with MeOH (entry 7)?

Answer: We appreciate reviewer 1 for the favorable comments. It should be 1,3-DCP, and we have corrected the error in our revised manuscript. **3a** was delivered in 85% yield with 86% e.e. with MeOH as an alcohol additive.

3. A model for the observed stereochemistry of allenylation is suggested.

Answer: As suggested by reviewer 3, a plausible model of the transition state of allenylation is outlined in our revised manuscript (Fig. 7). Initially, the proposed catalytic cycle for the generation of allenylphosphine oxide **3** begins with the Ni-mediated decarboxylation of propargylic carbonate **1**, and further delivers the allenylnickel complex **I**. Nucleophilic addition of the hydroxyphosphine anion **2'** onto the phosphine **L23**-bound allenylnickel intermediate **I** provide a stereoselective route to access chiral propargyl phosphinate **III**, and the following [2,3]-sigmatropic rearrangement furnishes the final allenylphosphine oxide **3**. We have included the model in our revised manuscript (Fig. 7).

REVIEWERS' COMMENTS

Reviewer #1 (Remarks to the Author):

In the revised version, author addressed my concerned problems commendably. I'm glad to recommend it publication in NC as current edition.

Reviewer #3 (Remarks to the Author):

The authors have completely and well addressed the reviewers' concerns. The present research have the novelty which should be meaningful to advance the catalytic propargylic transformation. The manuscript is well organized, and therefore recommended for the publication in Nat. Commun.